# Philadelphia-Negative MPN: A Molecular Journey, from Hematopoietic Stem Cell to Clinical Features

**DOI:** 10.3390/medicina57101043

**Published:** 2021-09-30

**Authors:** Valentina Giai, Carolina Secreto, Roberto Freilone, Patrizia Pregno

**Affiliations:** Division of Hematology, Città della Salute e della Scienza, 10100 Turin, Italy; csecreto@cittadellasalute.to.it (C.S.); rofreilone@cittadellasalute.to.it (R.F.); ppregno@cittadellasalute.to.it (P.P.)

**Keywords:** MPN, molecular landscape, hematopoietic stem cell, clinical, Philadelphia-negative

## Abstract

Philadelphia negative Myeloproliferative Neoplasms (MPN) are a heterogeneous group of hematopoietic stem cell diseases. MPNs show different risk grades of thrombotic complications and acute myeloid leukemia evolution. In the last couple of decades, from JAK2 mutation detection in 2005 to the newer molecular trademarks studied through next generation sequencing, we are learning to approach MPNs from a deeper perspective. Here, we intend to elucidate the important factors affecting MPN clonal advantage and the reasons why some patients progress to more aggressive disease. Understanding these mechanisms is the key to developing new treatment approaches and targeted therapies for MPN patients.

## 1. Introduction

Philadelphia-negative Myeloproliferative Neoplasms (MPN) are heterogeneous hematopoietic stem cell clonal diseases, clinically characterized by an increase of mature hematopoietic peripheral blood cells. According to the 2016 WHO classification [1], MPNs are grouped in polycythemia vera (PV), essential thrombocytopenia (ET), and myelofibrosis (MF). Other MPNs are chronic neutrophilic leukemia, chronic eosinophilic leukemia not otherwise specified, and MPN unclassifiable. From an epidemiological point of view, PV, ET, and MF are more frequent [2]. Compared to other myeloid malignancies, PV, ET, and MF have a more chronic course, with prognosis of decades for PV and ET and years for MF. Several prognostic scores have been developed for MPN: regarding ET and PV, age, JAK2-V617F mutation status, and recurrence of thrombosis had been the most important prognostic factors. In MF, several scores had been studied: before next generation sequencing (NGS) era, Dynamic International Prognostic Score System (DIPSS), and DIPSS-plus were the most used in clinical practice [3].

MPNs are chronic stem cell diseases. The only curative approach is allogeneic stem cell transplantation (ASCT). In PV and ET, usually ASCT is not an option, since mortality risks of ASCT are excessive compared to the relatively good prognosis of PV and ET. In young and performant MF patients, depending on prognostic scores, ASCT should be considered [4].

In the last years, due to the more available modern molecular biology techniques, such as NGS and Whole Genome Sequencing (WGS), a large number of mutations had been discovered in myeloid malignancies as well as in MPNs, contributing to elucidating new factors influencing the pathogenesis and evolution of these diseases. In this review, we intend to take a deeper look into the genetic landscape of MPNs starting from the hematopoietic stem cell (HSC) compartment to understand how this molecular characterization can ameliorate prognosis and treatment of MPN patients.

## 2. MPN Hematopoietic Stem Cells

HSCs reside in specialized microenvironments in the bone marrow called stem cell niches. The quiescent HSC is located in the endosteum, innervated by the sympathetic nervous system and irrorated by tiny arterioles. The more proliferating counterpart that gives rise to progenitors and more mature hematopoietic cells is settled close to the sinusoids, where macrophages promote erythroblasts growth and thrombopoietin improves DNA synthesis [5,6]. The endosteum is innervated by the sympathetic nervous system and, here, the HSC is tied to osteoblasts, via adherence proteins and thrombopoietin receptors. HSC quiescence is regulated by CXCL-4 and transforming growth factor beta 1 (TGF-β1) secreted by niche macrophages [7,8]. 

PV HSC are capable of inhibiting the growth of a normal stem cell compartment [9], enhancing bone marrow fibrosis, and disrupting sympathetic innervation in the stem cell niche, promoting myeloproliferation [8]. Additionally, normal HSCs are stimulated by PV HSC to release inflammatory cytokines. All these mechanisms are increased by age-derived microenvironment alterations [10] and may contribute to the clonal advantage of MPN stem cells [11].

In MF, the myeloproliferation is due to an intrinsic increase of thrombopoietin receptors [12], independently from JAK2, calreticulin (CALR), and thrombopoietin (TPO) receptor (MPL) mutational status [13]. Bone marrow fibrosis is reversible, and the increase of bone marrow function is due to a transformation of the MF HSC [14]. About 15–20% of PV patients evolve in MF [3].

In MPN, the primary JAK2-V617F mutation appears at a stem cell level [15]. When MPN HSC were transplanted to NOD SCID mice, the animals developed an MPN [16]. It was shown that the difference between PV and MF is the expansion of MPN HSC, which is higher in MF samples [15]. The JAK2-V617F allele burden is similar among the different MPNs, and variant allele frequencies (VAF) in neutrophils is comparable to HSC VAF [15]. Also, it was shown that the JAK2-V617F stem and progenitor cell compartment is not expanded in size; however, since JAK2 is more and more expressed in mature cells, the mutation causes an expansion of the terminal differentiated compartment [17]. From a therapeutic point of view, this is an important finding: it means that anti-JAK2-V617F treatments likely cannot eradicate the MPN stem cell compartment. On the contrary, the treatment with pegylated interferon α (peg-IFNα) can significantly decrease the clone size, and longer term remissions are possible [18].

MPNs are stem cell diseases: clonal evolution in acute myeloid leukemia occurs if additional mutations appear, such as ASXL1, TET2, DNMT3A, SF3B1, or SFSR2. 

Looking at the hematopoietic maturation tree, HSCs give rise to multipotent progenitors (MPP) that differentiate in common myeloid progenitors (CMP) or common lymphoid progenitors (CLP). Then, from CMP arise granulocyte macrophages progenitors (GMP) and megakaryocyte erythrocyte progenitors (MEP). GMP and MEP are precursors of granulocytes, macrophages, megakaryocytes, and erythrocytes [19,20]. Drivers mutations as JAK2-V617F, CALR, and MPL occur in the hematopoietic stem cells, causing an MPN phenotype or, in presence of another somatic mutations, to a malignant clonal evolution and progression (Figure 1). Clonal expansion of the lymphoid compartment is not conspicuous, because JAK2 is more and more expressed in the myeloid lineage and increased in the more mature population [17]. 

## 3. Molecular Pathogenesis

In the last years, progress has been made in the knowledge of MPN pathogenesis. Above all, many tyrosine kinases involved in the proliferation pathways of hematopoietic cells were identified.

In the initiation of the MPN clone, the JAK-STAT signalling pathway plays a central role. This pathway is activated by the extracellular binding of erythropoietin (EPO), TPO, and granulocyte-stimulating factors (G-CSF) to surface receptors on myeloid and erythroid cells. These receptors are associated with cytoplasmic tyrosine kinases of the Janus (JAK) family, including JAK2. When stimulated by a ligand, JAK kinases phosphorylate the signal transducer activators of transcription (STAT) proteins. When activated, STAT mediators migrate in the nucleus, stimulating gene transcription of proteins involved in proliferation. Several STAT proteins have been identified [21]; among these, the ones mainly related to hematopoietic cells growth are STAT5 (leading to a PV phenotype) [22] and STAT1 (inducing a ET phenotype) [23].

Clinical features of PV, ET, and MF are due to a disruption of balance between the intrinsic myeloproliferative stimulus of myeloid cells and the uncontrolled activation of the JAK-STAT pathway. 

Almost all cases of PV and at least half of the cases of ET and MF carry a driver mutation on the JAK2 gene on chromosome 9 (9p). In more than 95% of PV, the acquired replacement of a valine with a phenylalanine on codon 617 (V617F) in exon 14 of JAK2 leads to a resistance of the kinase domain (JH1) to the regulatory and inhibitory control of the JH2-domain; as a consequence, the JH1-domain is constitutively active. Mutations can occur in heterozygosis, where the presence of extracellular ligands is necessary to activate the JAK-STAT cascade, or in homozygosis, where cell proliferation is independent from external stimulus [24]. In less than 5% of patients with PV, the mutation occurs in JAK2 exon 12; PV with exon 12 mutation shows a less severe erythrocytosis compared to JAK2-V617F mutated PV [25]. The different clinical presentations dependent on JAK2 mutations of PV, ET, and MF could be partially explained by the different allele burden of JAK2; for example, when a high JAK2-V617F allele burden is expressed, the disease shows a more aggressive behaviour [26].

Other than JAK2, the direct or indirect activation of MPL can lead to an ET and MF clinical phenotype. The mutation of the CALR gene on exon 9 of chromosome 19 is found in up to 25% of cases of ET or MF; this mutation indirectly stimulates the JAK-STAT pathway, increasing MPL activity in hematopoietic stem cells with consequent thrombocytosis. Namely, the CALR gene encodes molecules which are sited in the endoplasmic reticulum and that regulate the correct protein folding and distribution in the cell. Mutated CALR encodes proteins with an aberrant C-terminus, which activates the MPL receptor and, thus, proliferation through JAK-STAT cascade. Two main mutations have been identified in CALR gene: deletions define the type 1 mutation, while insertions define type 2 mutations. The first is more common in primary MF [27].

Finally, a small percentage of ET and MF (5–7%) directly carries mutations in the TPO receptor (MPL) at W515 or S505 on exon 10 of chromosome 1p, which makes it constitutively active, in some cases even in the absence of TPO. As the indirect stimulation on the JAK-STAT pathway by CALR mutation, the MPL mutation on hematopoietic stem cells leads to extreme thrombocytosis [28,29,30]. 

JAK2, CALR, and MPL mutations are mutually exclusive in up to 50% of patients with MPN. Thus, taken together, almost all cases of Philadelphia-negative MPN are explained by one of these three driver mutations. Still, a classic driver mutation is not detected in up to 10% of patients with ET or MF, defined as “triple negative” [31]. 

In addition to the JAK-STAT pathway, other molecular pathways are involved in MPN pathogenesis. Mutations in JAK2, CALR, and MPL can activate phosphoinositide 3-kinase (PI3K) and mitogen-activated protein kinase (MAPK) pathways, which transduce proliferation signalling through complex mechanisms of kinase phosphorylation [32,33]. The MAP kinase (also known as Erk) is activated by the monomeric GTPase Ras, which starts a cascade signalling through other connection kinases (namely, Raf and Mek), priming the transcription of nuclear cyclins [34]. Similarly, cellular growth and survival are stimulated by the activation of the serine/threonine kinase mTOR through the PI3K-Akt signalling [35]. Further studies are required to gain a deeper understanding of the overactive signalling cascade: the combination of therapies targeting different pathways could result in a more efficient blockage of uncontrolled proliferation.

NGS tests are modern molecular biology techniques able to sequence cell DNA and improve genomic research. In the last decade, NGS has been widely used in hematology to better understand molecular pathways involved in leukemogenesis. The other side of the coin is the generation of an enormous amount of molecular data, sometimes difficult to understand. As a result, a huge number of studies focused on analyzing whether certain mutations can influence the clinical outcome and prognosis in hematological malignancies and also in MPN [36]. 

Studies showed that gene expression in neutrophils in patients with MPN differs from gene expression in neutrophils of the normal population. No differences in gene expression were found between PV, ET, and MF [37]. The involved genes regarded cytokines and growth factors, such as interleukin-6, interleukin-10, interleukin-8, granulocyte-macrophages colony stimulating factors, and transforming growth factor beta (TGF-b) [38]. Furthermore, looking at the HSC compartment, it was shown that MPN gene expression differed from the healthy counterpart, but also among PV, ET, and MF HSC. These data suggest that PV, ET, and MF are genetically three different diseases [9,39,40,41]. 

### 3.1. Germline Predisposition

Constitutional variation in genes that can influence the MPN onset can be distinguished in two categories: (1) mutations that occur in the whole population and are responsible for a small increase in MPN, and (2) familial mutations with high penetrance that can increase the risk from 1.5 to 3 of MPN development, like TERT or the JAK2 46/1 haplotype [42]. Germline genetic background can influence the emerging MPN and its manifestation, being the fertile soil where the phenotypic driver mutation can come up and launch the disease. Genes involved in cells senescence, such as TERT in JAK-STAT signalling; SH2B3 in myeloid differentiation; GFI1B in DNA damage and repair; ATM and CHEK2 in epigenetic regulators; and TET2 were identified as targets of several additional predisposition loci in MPN onset [43].

### 3.2. Additional Somatic Mutations

One third of patients with MPN displays at least one additional somatic mutation. These mutations affect genes involved in epigenetic regulation (TET2, DNMT3A, IDH1/2), chromatin modification (ASXL1, EZH2, IDH1/2), in the splicing machinery (U2AF1, SF3B1, SRSF2, ZRZS2), and DNA repair (Tp53) [27].

ASXL1, DNMT3A, and TET2 are quite common and displayed in more than 5% of patients. Others, such as CBL, SF3B1, EZH2, TP53, SRSF2, USAF1, and IDH1/2 are present in less than 2% of MPN patients [36].

Several studies investigated how additional mutations could influence clinical features of the disease. Type and number are associated with phenotype and prognosis. For example, mutated NFE2 is related to PV, while mutations of the splicing machinery genes, such as SF3B1 and SRSF2, are involved in increased fibrosis in MF [44]. Others, such as IKZF1, are linked to blast crisis evolution [36,45,46].

DNA methylation is a biological process needed to control and regulate HSC senescence and differentiation [47]. DNMT3A is a methyl transferase, and TET2 encodes for a protein involved in demethylation. Somatic mutations in these two genes alter cell differentiation and proliferation. The depletion of DNMT3A changes stem cell function [48]. In MPN, both genes are mutated and do not work (as loss of function). Mutated TET2 can promote and inhibit HSC differentiation, and loss of DNMT3A function could lead to transformation in acute myeloid leukemia (AML) [47]. IDH1/2 is a gene involved in methylation and DNA damage: knock-in mice for IDH1/2 showed higher hematopoietic stem and progenitor cells (HSPC) proliferation, intense anemia, and extramedullary disease. IDH1/2 is mutated in around 1% of MPN patients [36] and it is considered a high risk mutation in MF [49].

Another gene involved in methylation and regulation of histones metabolism is EZH2. EZH2-related loss of function brings to deregulation of HSC self-renewal, to enhance fibrosis and to reduce erythropoiesis in the JAK2 V617F mutated environment [50]. 

Many studies investigated mutation prognostic roles in MF. Some mutations are linked to an inverse outcome; for example, mutations of Tp53, IDH1/2, and SRSF2 confer an increased risk of leukemia evolution and mutations of ASXL1, EZH2, and SFSR2 are linked to shorter overall survival (OS) [14,51,52]. Recently, it was shown that ASXL1 mutation alone does not impact the outcome; however, it does when associated to Tp53 or to EZH2, CBL, U2AF1, SRSF2, IDH1, IDH2, NRAS, or KRAS. On the contrary, Tp53 mutation alone heavily affects leukemia transformation and death [49]. ASXL1 is mutated in 25% of MF patients [53,54] (Table 1).

In a large cohort of 2035 MPN patients, 45% of patients showed just JAK2, MPL, and CALR mutations, while 5 patients displayed 33 driver mutations. The number of driver mutations increased with disease stage and age. Eight subgroups of MPN were identified, based on the genomic characterization with diverse prognosis [36]. In the last years, many prognostic scores combining together clinical features with molecular data were developed for MPN [63,64,65,66,67,68] (Table 2 and Table 3). 

## 4. Clinical Implications

Ruxolitinib is a selective JAK1/2 inhibitor that showed clinical benefits in MF and PV refractory to standard therapies. In the phase 3 trial by Verstovsek et al., 309 patients affected by intermediate-2/high risk MF were randomly assigned to placebo or ruxolitinib. The latter was associated with better symptom control (46% vs. 5%), a reduction in spleen size by 35% or more (42% vs. 0.7%), and improved OS, with acceptable haematologic toxicity (mainly anemia and thrombocytopenia) [69]. In the ruxolitinib group, 72% of patients were JAK2-V617F positive; among responders, clinical efficacy was higher in patients with the mutation compared to JAK2-V617F negative ones (reduce in spleen size 34.6% vs. 23.8%; improvement in symptoms score 52.6% vs. 28%). Despite this, the effectiveness of ruxolitinib was seen independently from JAK2 mutational status, as confirmed by some studies [17,70]. 

Regarding PV, Vannucchi et al. run a phase 3 randomized trial on 222 patients comparing the efficacy of ruxolitinib versus standard care in patients intolerant/refractory to first line therapy with hydroxyurea. In addition in this case, therapy with ruxolitinib was effective in improving symptoms and spleen size compared to placebo; almost 24% of patient reached a complete hematologic response (defined as hematocrit, platelets and white-cell count control) [71].

In both studies, the allele burden of JAK-V617F decreased during ruxolitinib treatment compared to controlled arms. Unfortunately, VAF reduction did not conduct to a prognostic improvement. 

Ruxolitinib-resistant or intolerant patients show an inferior OS and an increased risk of AML evolution [72]. From a molecular view, these patients acquire additional mutations, such as ASXL1, TET2, EZH2, and Tp53 [73]. The same study by Newberry et al. showed that OS of resistant or intolerant MF patients was shorter in the case of AML progression [74]; treatment with ruxolitinib did not increase the risk of AML evolution, compared to hydroxyurea therapy [75]. Patients that acquired additional mutations under uxolitinib therapy showed a shorter OS in both resistant or intolerant groups. In the last years, new strategies have been developed for both ruxolitinib-naïve and resistant patients. Fedratinib is a new selective JAK2 and FLT3 inhibitor approved in 2019 in the United States for the treatment of intermediate 2/high risk MF in patients with platelets more than 50000/μL, both with prior exposure to ruxolitinib or not [76]. In the randomized phase III trial from Pardanani et al., fedratinib resulted in a significant response in terms of improved symptoms burden and decreased spleen volume more than 35% compared to placebo [77]. The most common haematological adverse events were anemia and thrombocytopenia, but only a few patients permanently discontinued the study drug due to these complications. However, fedratinib is not yet available in Europe outside of clinical trials [78]. Two other drugs are under investigation for the treatment of high-risk MF, namely pacritinib and momelotinib. The first is a JAK2 inhibitor that showed a superior response in terms of reducing spleen volume and symptom burden compared to the best available therapy (in most cases, ruxolitinib) in patients with MF and thrombocytopenia [79]; the latter is a JAK1/2 inhibitor that showed a response in reducing transfusion burden, but less activity in reducing spleen size compared to the best available therapy (7% vs. 6%) in patients previously treated with ruxolitinib [80]. Another new drug under investigation is imetelstat, a telomerase inhibitor that showed promising results in terms of OS and symptoms burden in ruxolitinib-resistant MF in a phase 2 study [81,82].

Hematopoietic stem cell transplantation still represents the only curative treatment for high risk MF (namely, patients with DIPSS intermediate 2/high risk, high transfusion burden, adverse cytogenetic), whose benefits should be balanced with the morbidity and mortality associated with the procedure [3].

In some cases, studies showed that a certain MPN molecular profile can suggest a treatment. For example, JAK2-V617F+ U2SFR+ SF3B1+ ASXL1- high/intermediate-2 MF showed a superior overall response rate when treated with imetelstat [83]. In PV patients, the presence of additional mutations is associated with a lower decrease of JAK-V617F allele burden during peg-IFNα treatment [84]. Also, peg-IFNα cannot eradicate PV TET2+ clone. Similarly, in CALR+ ET, the CALR allele burden did not significantly decrease when cells show TET2, IDH2, ASXL1, and Tp53 mutations [18].

The molecular characterization of MPN gives new insights to understand how the MPN stem cell starts the clonal advantage and, via other mutations, initiates the malignant progression. Further studies are needed to better understand these heterogenous and jeopardized diseases. The molecular characterization must help researchers developing new targeted drugs to prolong OS, ameliorate prognosis, and improve quality of life.

## Figures and Tables

**Figure 1 medicina-57-01043-f001:**
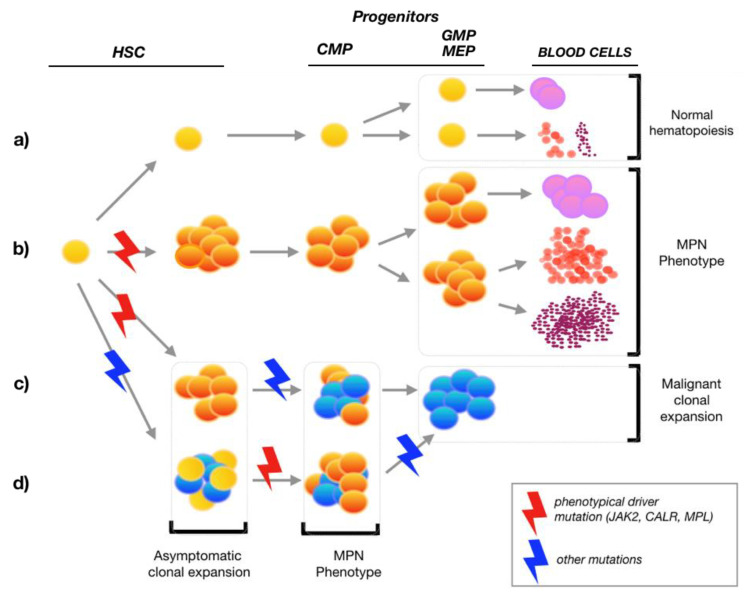
From hematopoietic stem cell (HSC) to clonal evolution. In this figure, we show the development of four possible scenarios in MPN pathogenesis: (**a**) the normal HSC that generates normal myeloid progenitors and blood cells (granulocytes in violet, erythrocytes in light red, and platelets in dark red); (**b**) a phenotypical driver mutation causing the expansion of MPN progenitors and increased blood cells count; (**c**) an additional somatic mutation, occurred after an initial starting driver mutation, leading to malignant clone development, as MPN secondary acute myeloid leukemia (AML); (**d**) in this case, the starting mutation is not a classic phenotypical driver mutation (JAK2, CALR, or MPL), but another, leading to an initial asymptomatic clonal expansion; on this population, then, the JAK2, CALR, or MPL occurs and the malignant clone expansion begins. One mutation is not sufficient for the development of an aggressive disease. *Abbreviations*: CMP, common myeloid progenitors; GMP, granulocyte macrophages progenitors; MEP, megakaryocyte erythrocyte progenitors.

**Table 1 medicina-57-01043-t001:** Current main mutations in MPN that influence outcomes and clinical features.

Category	Gene	Function/Mutation Effect	Effects on Prognosis	REF	
Histone modification	ASXL1	Demethylation and transcription repression by heterozygous mutations.	Increased AML evolution and fibrosis development.	[49,51,55]	
Histone modification	EZH2	Histone methyltransferase and transcription repression by heterozygous and homozygous mutation.	Increased AML evolution and fibrosis development.	[46,56]	
DNA Methylation regulation	DNMT3A	Reduced methyltransferase activity in DNA and histone methylation.	Reduced OS in MF.	[55,57]	
DNA Methylation regulation	IDH1/2	Epigenetic dysregulation influencing leukemogenesis. Heterozygous mutation.	Reduced OS in MF.	[14,58]	
Splicing machinery	SRSF2	Needed for splicing of pre-mRNA. Heterozygous mutations.	Reduced OS in MPN and increased risk of AML evolution.	[44,51]	
Splicing machinery	U2AF1	Needed for splicing of pre-mRNA. Heterozygous mutations.	Disease progression and reduced OS in MF.	[59,60]	
Signalling	CBL	Increased STAT5 signalling. Homozygous mutations.	Reduced OS in MF. Resistance to JAK inhibitors.	[61,62]	
Signalling	NRAS/KRAS	Increased proliferation. Heterozygous mutations.	Reduced OS in MF. Resistance to JAK inhibitors.	[57,61]	
Signalling	PTPNI1	Activation of signalling.	Reduced OS in blastic phase.	[57]	
Transcription	RUNX1	Role in regulation of normal hematopoiesis.	Reduced OS in blastic phase.	[36]	
Transcription	TP53	Regulation of apoptosis and cell cycle arrest.	Reduced OS in MPN and increase of disease progression.	[57,61]	
Splicing machinery	SF3B1	Member of the splicing machinery.	Increased risk of fibrotic evolution.	[55]	

In red, mutations impacting negatively on survival. In yellow, the SF3B1 mutation that increases bone marrow fibrosis, but not the prognosis. Abbreviation: OS, overall survival; AML, acute myeloid leukemia; MF, myelofibrosis.

**Table 2 medicina-57-01043-t002:** Clinical-molecular prognostic scores for PV and ET.

Prognostic Score	Variables (Points)	Points	Risk Categories (Points)	Median Survival (Years)
MIPSS-PV [63]	Leukocyte count ≥ 15 × 10^9^/L Thrombosis history Age > 67 years SRSF2 mutation	1 1 2 3	Low (0–1) Intermediate (2–3) High (4–7)	24 13.1 3.2
MIPSS-ET [63]	Leukocyte count ≥ 11 × 10^9^/L Age > 60 years Male sex SRSF2, SF3B1, U2AF1 and TP53 mutation	1 4 1 2	Low (0–1) Intermediate (2–5) High (6–8)	34.3 14.1 7.9

Abbreviations: MIPSS, Mutation-Enhanced International Prognostic Scoring System; PV, polycythemia vera; ET, essential thrombocythemia.

**Table 3 medicina-57-01043-t003:** Clinical molecular prognosis scores for myelofibrosis.

Prognostic Score	Variables (Points)	Points	Risk Categories (Points)	Median Survival (Years)
MIPSS70 [64]	Hemoglobin < 10 g/dL Blasts > 2% Constitutional symptoms Leukocytes > 25 × 10^9^/L Platelet count < 100 × 10^9^/L BM fibrosis ≥ 2 Non CALR type-1 HMR = 1 HMR ≥ 2	1 1 1 2 2 1 1 1 2	Low (0–1) Intermediate (2–4) High (5–12)	27.7 7.1 2.3
MIPSS70 plus [64]	Hemoglobin < 10 g/dL Blasts > 2% Constitutional symptoms Non CALR type-1 HMR = 1 HMR ≥ 2 Unfavourable karyotype	1 1 1 2 1 2 3	Low (0–2) Intermediate (3) High (4–6) Very high (7–11)	20.0 6.3 3.9 1.7
MIPSS70 plus v2.0 [65]	Hemoglobin 8–10 g/dL Hemoglobin < 8 g/dL Blasts > 2% Constitutional symptoms Non CALR type-1 HMR+U2AF1 Q157 = 1 HMR+U2AF1 Q157 ≥ 2 HR Karyotype VHR Karyotype	1 2 1 2 2 2 3 3 4	Very low (0) Low (1–2) Intermediate (3–4) High (5–8) Very high (9–14)	Not reached 10.3 7 3.5 1.8
GIPSS [66]	Non CALR type-1 ASXL1 mutation SRSF2 mutation U2AF1 Q157 HR karyotype VHR karyotype	1 1 1 1 1 2	Low (0) Intermediate-1 (1) Intermediate-2 (2) High (3–6)	26.4 8.0 4.2 2.0
MYSEC-PM [67]	Hemoglobin < 11 g/dL Blasts ≥ 3% Platelets < 150 × 10^9^/L Constitutional symptoms Age at secondary MF (0.15 point/year) CALR not mutated genotype	1 1 1 2 2	Low (<11) Intermediate-1 (11-14) Intermediate-2 (14-16) High (≥16)	Not reached 9.3 4.4 2.0
MTSS [68]	Platelets < 150 × 10^9^/L Leukocytes > 25 × 10^9^/L Karnofsky PS < 90% Age ≥ 57 years HLA-mismatched unrelated donor Non CALR/MPL mutation ASXL1 mutation	1 1 1 1 2 2 1	Low (0–2) Intermediate (3–4) High (5) Very high (6–9)	5-years OS 83% 5-years OS 64% 5-years OS 37% 5-years OS 22%

Abbreviations: MIPSS, Mutation-Enhanced International Prognostic Scoring System; GIPSS, Genetically Inspired Prognostic Scoring System; MYSEC-PM, Myelobrosis Secondary to PV and ET-Prognostic Model; MTSS, Myelobrosis Transplant Scoring System; OS, overall survival; BM, bone marrow; PS, performance status. High molecular risk (HMR): ASXL1, SRSF2, EZH2, IDH1/2. Unfavourable karyotype: any abnormal karyotype other than normal karyotype or single abnormalities of 20q2, 13q2, +9, chromosome 1 translocation/duplication, -Y, or sex chromosome abnormality other than -Y. High risk (HR) karyotype: all the abnormalities that are not VHR and favourable (normal karyotype or single abnormalities of 20q−, 13q−, +9, chromosome 1 translocation/duplication, or sex chromosome abnormality including -Y). Very high risk (VHR): single or multiple abnormalities of −7, inv (3), i(17q), 12p−, 11q−, and autosomal trisomies other than +8 or +9.

## Data Availability

Data supporting reported results can be found on www.pubmed.com.

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
