# Peer review of "Philadelphia-Negative MPN: A Molecular Journey, from Hematopoietic Stem Cell to Clinical Features"

_medicina, 2021, doi:10.3390/medicina57101043_

Round 1
Reviewer 1 Report
In this review, Giai et al. summarized the available biolocial and molecular data on the effect of different molecular alteration that may promote the clonal advantage that characterize MPN. They reported the various molecular alterations linked to the uncontrolled activation of the JAK-STAT pathway in PV, TE and MF. The second part of the manuscript nicely summarize the main mutations that has been found to influence the outcome of MPN patients. In the last part of the manuscript, the authros reported the role of HSC and clonal evolution in MPN.. The review is interesting, well-structured and comprehensive and contains all important aspects and landmark papers. The Tables of the manuscript are very complete and well designed.
Author Response
Thank you for the revision and comments.
Reviewer 2 Report
The review article with the title "Philadelphia-negative MPN: a molecular journey, from hematopoietic stem cell to clinical features" is an interesting article. The authors present important mutations which affect the prognosis of patients with Philadelphia-negative myeloproliferative neoplasms (MPN), as well as some prognostic scores for MPN patients and specific treatment strategies which could be adopted for these patients. Even though it is an informative article, there are a few issues, as explained below:
- The parts about prognosis and potential treatment strategies for patients with MPN are limited. The authors are advised to include more relative information.
- The structure of the manuscript may confuse the reader. First, it is advised to include the part about the hematopoietic cell maturation in the beginning of the manuscript, together with more information about these diseases. The subtitle "JAK-STAT pathway" is confusing as the authors do not explain only the mechanism of this pathway and they want to show the association of mutations in genes which belong to this pathway with MPN. Moreover, this part could be presented possibly after the germline and the somatic mutations.
- The language of the manuscript could be improved, as many mistakes can be found.
- The authors state "The only curative approach is allogeneic stem cell transplantation (ASCT); in PV and ET, usually ASCT is not an option". Why it is not an option?
- What do the authors mean by "Still, a classic driver mutation is not detected in up to 10% of patients with a clinical phenotype of ET or MF, defined as “triple negative”. What is this phenotype?
Author Response
POINT 1: The parts about prognosis and potential treatment strategies for patients with MPN are limited. The authors are advised to include more relative information.
Response point 1: We added a part in the MPN treatment (in red in the text)
POINT 2: The structure of the manuscript may confuse the reader. First, it is advised to include the part about the hematopoietic cell maturation in the beginning of the manuscript, together with more information about these diseases. The subtitle "JAK-STAT pathway" is confusing as the authors do not explain only the mechanism of this pathway and they want to show the association of mutations in genes which belong to this pathway with MPN. Moreover, this part could be presented possibly after the germline and the somatic mutations.
Response point 2: We moved the hematopoietic stem cell part before the “molecular” part. We changed titles.
POINT 3: The language of the manuscript could be improved, as many mistakes can be found.
Response point 3: We corrected the mistakes
POINT 4: The authors state "The only curative approach is allogeneic stem cell transplantation (ASCT); in PV and ET, usually ASCT is not an option". Why it is not an option?
Response point 4:
We changed the sentence: in PV and ET, usually ASCT is not an option, since mortality risks of ASCT are excessive compared to the relative good prognosis of PV and ET.
POINT 5: What do the authors mean by "Still, a classic driver mutation is not detected in up to 10% of patients with a clinical phenotype of ET or MF, defined as “triple negative”. What is this phenotype?
Response point 5: We changed the sentence, deleting “clinical phenotype” that could be confusing.
Reviewer 3 Report
minor corrections needed. see mauscript

Author Response
Dear Reviewer,
we corrected the abbreviations and we accepted your advices in the text.